# A cost description of the setup costs of community-owned maternity waiting homes in rural Zambia

**Allison Juntunen**[1☯]*, **Nancy A. Scott**[1☯], **Jeanette L. Kaiser**[1], **Taryn Vian**[2], **Thandiwe Ngoma**[3], **Kaluba K. Mataka**[4], **Misheck Bwalya**[3], **Viviane Sakanga**[5], **David Kalaba**[6], **Godfrey Biemba**[7], **Peter C. Rockers**[1], **Davidson H. Hamer**[1,8], **Lawrence C. Long**[1]

**1** Department of Global Health, Boston University School of Public Health, Boston, MA, United States of America, **2** School of Nursing and Health Professions, University of San Francisco, San Francisco, CA, United States of America, **3** Department of Research, Right to Care Zambia, Lusaka, Zambia, **4** Zenysis Technologies, Lusaka, Zambia, **5** Amref Health Africa in Zambia, Lusaka, Zambia, **6** Eastern and Southern African Management Institute (ESAMI), Arusha, Tanzania, **7** National Health Research Authority, Pediatric Centre of Excellence, Lusaka, Zambia, **8** Section of Infectious Diseases, Department of Medicine, Boston University School of Medicine, Boston, MA, United States of America

☯ These authors contributed equally to this work.

* juntunen@bu.edu

**Data Availability Statement:** Data used to generate the analysis in this manuscript may be accessed through OpenBU, an open access data repository. Two datasets were used and can be

## Abstract

Maternity waiting homes (MWHs) are one strategy to improve access to skilled obstetric care in low resource settings such as Zambia. The Maternity Homes Access in Zambia project built 10 MWHs at rural health centers in Zambia for women awaiting delivery and postnatal care (PNC) visits. The objective of this paper is to summarize the costs associated with setup of 10 MWHs, including infrastructure, furnishing, stakeholder engagement, and activities to build the capacity of local communities to govern MWHs. We do not present operational costs after setup was complete. We used a retrospective, top-down program costing approach. We reviewed study documentation to compile planned and actual costs by site. All costs were annuitized using a 3% discount rate and organized by cost categories: (1) Capital: infrastructure and furnishing, and (2) Installation: capacity building activities and stakeholder engagement. We assumed lifespans of 30 years for infrastructure; 5 years for furnishings; and 3 years for installation activities. Annuitized costs were used to estimate cost per night stayed and per visit for delivery and PNC-related stays. We also modeled theoretical utilization and cost scenarios. The average setup cost of one MWH was $85,284 (capital: 76%; installation: 24%). Annuitized setup cost per MWH was USD$12,516 per year. At an observed occupancy rate of 39%, setup cost per visit to the MWH was USD$70, while setup cost per night stayed was USD$6. The cost of stakeholder engagement activities was underbudgeted by half at the beginning of this project. This analysis serves as a planning resource for governments and implementers that are considering MWHs as a component of their overall maternal and child health strategy. Planning considerations should include the annuitized cost, value of capacity building and stakeholder engagement, and that cost per bed night and visit are dependent upon utilization.

found in the repository: 1) the costing database, which details all purchases made and corresponding cost-code, and 2) the maternity waiting home register, from which we reported MWH occupancy and women's average length of stay. The following link leads to our datasets: https://hdl.handle.net/2144/45243.

**Funding:** This program was developed and implemented in collaboration with Merck for Mothers, Merck's 10-year, $500 million initiative to help create a world where no woman dies giving life. Merck for Mothers is known as MSD for Mothers outside the United States and Canada (MRK 1846-06500.COL to NAS). The development of this article was additionally supported in part by the Bill & Melinda Gates Foundation (OPP1130329 to NAS) https://www.gatesfoundation.org/How-We-Work/Quick-Links/Grants-Database/Grants/2015/07/OPP1130329 and The ELMA Foundation (ELMA-15-F0017 to NAS) http://www.elmaphilanthropies.org/the-elma-foundation/. LL was supported by the National Institute of Mental Health of the National Institutes of Health (Award Number K01MH119923 to LL). The funders had no role in study design, data collection and analysis, decision to publish, or preparation of the manuscript. The content is solely the responsibility of the authors and does not necessarily reflect positions or policies of Merck for Mothers, the Bill & Melinda Gates Foundation, NIH, or The ELMA Foundation. Funding for Online Open publication supported by Chronos Support through the Bill & Melinda Gates Foundation.

**Competing interests:** The authors have declared that no competing interests exist.

## Introduction

Maternity Waiting Homes (MWHs), residential facilities located near equipped health facilities or hospitals, are a long-used strategy to address challenges in reaching skilled obstetric care services in rural and remote areas of low resource settings, such as Zambia [1–3]. Distance, time, and difficulty in finding reliable and affordable transportation are well-documented barriers to accessing obstetric care at an equipped health facility in these low resource settings [4–9], yet delivery at a health facility capable of providing basic emergency obstetric and neonatal care is known to improve maternal and neonatal health outcomes [10–13].

Researchers have found evidence that MWHs are associated with increased utilization of health facilities for delivery and improved maternal and neonatal health outcomes including reductions in obstetric complications and stillbirths [14]. While the quality of MWHs varies within and between countries, there is a growing consensus that designs developed through community participation and governance are more likely to be culturally appropriate and display features of safety and comfort for waiting women. Thus, they are more likely to be locally accepted and utilized [15]. However, few studies have reported the costs associated with implementing community-managed MWHs.

The Maternity Homes Access in Zambia (MAHMAZ) project, implemented by Right to Care Zambia and evaluated by Boston University, under the Maternity Homes Alliance (MHA), in collaboration with the government of the Republic of Zambia, has expanded the operations of MWHs in rural Zambia since 2015. Before constructing new MWHs, the MHA solicited feedback from a variety of national and local leaders, beneficiaries and stakeholders, from which the MHA developed a Core MWH Model, designed to be culturally appropriate, feasible, and sustainable through local ownership and operation, and projected utilization and costs [16, 17].

Understanding and planning setup costs of interventions is as important to informed decision making as understanding intervention effectiveness. Here we present a partial, top-down description of the direct financial costs required to: (1) construct 10 MWHs, (2) equip the MWHs with the necessary supplies and furnishings, (3) build the capacity of local community members to oversee and operate the MWHs, and (4) engage stakeholders to increase buy-in from the national to local levels. We compare actual costs to planned costs estimated prior the start of the MAHMAZ study and present the cost per visit to the MWH based on occupancy over a 12-month period. This article serves as a planning resource for governments or organizations considering implementing MWHs as an initiative in their broader maternal health strategy.

## Methods

### Study setting

MAHMAZ established the new MWHs in 4 rural districts: Choma, Kalomo, and Pemba in Southern Province, and Nyimba in Eastern Province [18]. These districts are primarily rural, poor, and have relatively low utilization of maternal health services [8]. The 10 sites were an average of 47.5 (SD 16.7) kilometers (km) from the closest urban center and 235 km (SD 39.6) from the capital city of Lusaka [9]. Many sites required travel along unpaved, dirt or gravel roads.

### Intervention description

MAHMAZ constructed and implemented 10 new MWHs over 42 months. Construction, spanning from preliminary consultations to final inspections, took place over 28 months (July

2015 to October 2017). Capacity building activities were held concurrently with construction, and a package or trainings and mentorship activities (S1 Table) were delivered sporadically between September 2015 to November 2017. Additionally, key stakeholders reconvened for final dissemination activities in December 2018 and January 2019. The Core MWH Model, designed through formative research and operationalized by MAHMAZ, included a set of criteria that each new MWH must meet [15, 19]. The Core MWH Model requires MWHs be formally linked to the existing local health system to ensure continuity of care for waiting women. To operationalize this, health facility staff participated in the oversight of the homes and checked in on waiting women routinely for changes in clinical condition, though did not provide formal clincal care in the MWH. MAHMAZ constructed its MWHs adjacent to existing rural health centers and less than 2 hours from referral hospitals capable of managing obstetric emergencies. The intention of the MWH was to provide lodging for women awaiting delivery for 10–14 days before their expected delivery date based on guidance from the WHO, the GRZ, and formative research in the region [15, 20]. This timeframe was meant to ensure early deliveries are not missed. Factors affecting length of stay and occupancy rates will be explored in future work. Waiting women were invited to participate in health education classes, conducted multiple times per week by health center staff and community-based volunteers. Consistent with government policy, the MWHs were designed to be free of charge for women and their companions awaiting antenatal care (ANC), labor and delivery, or postnatal care (PNC) services. MAHMAZ did not provide food for the waiting women as part of the initial support materials, though sites occasionally opted to use contributions from the surrounding community to feed the waiting women. Operational costs of running the MWHs were planned to be covered by income-generating activities and are not reported here. These activities were managed through social enterprises overseen by MWH governance committees. The MWH governance committees also created policies to guide and manage the MWHs. Table 1 summarizes the key elements related to MWH infrastructure, furnishing, capacity building activities necessary for governing and managing the MWHs, and key stakeholder engagement activities, as well as the associated budget line-items that were costed. S1 Table further describes the scope of the capacity building and stakeholder engagement activities. More detail on each of these activities is provided in the sections below.

**MWH infrastructure & furnishing.** *Building specifications*. MAHMAZ constructed 10 nearly identical MWHs, each with concrete foundation, floor, and walls, metal roof, and glass windows. Project personnel initially considered renovating existing MWHs with linkages to health facilities, rather than constructing new MWHs. Formative research, however, found existing homes in such poor condition that rennovations would be extensive and new construction would be more structurally sound. Additionally, in these remote areas, other types of accommodations, such as lodges or guest houses, were not readily available and formative research suggested MWHs were a feasible and acceptable option in this context [15]. Constructing new homes assured high quality structures and allowed the team to include elements that were important to traditional leadership, CHWs, and expectant mothers such as a private bathing area, separate space for postnatal women, and a large covered gathering space. Each MWH had a 10-bed dormitory for antenatal women, a 4-bed dormitory designated for postnatal women (as requested during formative research with the beneficiaries and relevant stakeholders [15]), a small office and storage room, a walled courtyard, 3 latrines, and 3 private bathing stalls. Only 1 site had access to piped water which was connected to sinks and toilets. All sites were wired for electricity, to be connected when the rural electrification project reached their areas. All sites were provided with 2 solar lights, the cost of which was included under furnishings. A small, covered cooking area was constructed next to each MWH. Each

**Table 1. Description of the budget line-items associated with each activity and the overarching costing category.**

| | Associated Activities | Associated Budget Line-Items |
|---|---|---|
| **Capital costs** | | |
| Infrastructure | Design of MWH structure | Architecture firm contract |
| | Site visits with companies tendering MWH bids | Per diems for project staff; transport and fuel for project staff |
| | Construction of 10 MWH buildings | Contractor contract (includes labor, construction materials, equipment, transportation) |
| | | Construction insurance |
| | | Materials & transport (additional construction materials purchased by project) |
| | Construction manager oversight trips | Per diems, accommodation, fuel, and transport for project staff |
| | Government oversight visits | Per diems, accommodation, fuel, and transport for project staff; per diems, fuel, and transport for ministry personnel |
| Furnishings | Purchase of furnishings for 10 MWHs | See S1 Table for a list of items purchased for each MWH |
| | Transport of furnishings to 10 MWHs | Per diems, accommodation, fuel, and transport for project staff; truck and driver hires, and fuel to transport materials |
| **Installation costs** | | |
| Capacity building | Trainings, workshops, supervision, mentorship for governance committees and management units (see Table 2) | Supplies, printing, and venue; food and drinks, per diems, accommodation, transport, and fuel for project staff and participants; consultant/trainer fees |
| Stakeholder engagement | National and site launches (see Table 2) | Supplies, printing, and venue; tent, chair, and audio equipment rentals; hire of local theatre groups; food, drinks, per diems, accommodation, transport, and fuel for project staff and participants; videographer |
| | Community-level engagement (see Table 2) | Supplies and printing; food, drinks, per diems, accommodation, transport, and fuel for project staff |

MWH had an attached open-air verandah with concrete benches as a multipurpose space for recreation or relaxation. Blueprints and photos of the MWHs can be found in S1 and S2 Figs.

*Construction.* A single MWH floor plan was designed by a local architecture firm (S1 Fig) and an independent contractor constructed the MWHs with oversight from project staff. The contractor was selected through a competitive sealed bid tender process. One company won the bid for the project which included all labor, materials, and transportation for the main MWH structures. Some communities donated labor and supplies, primarily sand, for which we were unable to reliably estimate costs. These contributions were encouraged for community buy-in but were considered negligible to the overall cost of the project.

*Furnishings.* Main furnishings included beds and mattresses for women, extra mattresses for companions (local practice is for a companion to accompany the woman), bedding, lockable cabinets, mosquito nets, solar lights, cooking supplies and office furniture. All furnishings were procured in-country after careful review of quotations from at least 3 vendors. A local trade school built the beds and office shelves. A local non-profit tailoring organization made the bed covers. Detailed information on structural elements and furnishings is provided in S2 Table.

**MWH capacity building: Governance & management systems.** All major activities conducted by project staff related to building the capacity of local communities to oversee the operations of the MWHs and stakeholder engagement and are included in the following sections and in S1 Table. All of the activities described were related to the setup of the MWH, establishment of governance committees and management units, and garnering preliminary stakeholder support. While some of these trainings and meetings may have recurred through the course of the project, we do not include those activities here. All activities included are related to setup, and we do not estimate running costs.

*Governance committees.* The Core MWH Model included a formalized governance and management system; financial records systems; standard operating procedures for intake, registration,

and monitoring; and a mechanism for feedback. Informed by health systems personnel and traditional leaders, MAHMAZ designed a 2-layer system of governance and management with representation from health facility staff, community-based volunteers, traditional leadership, and other community members [21]. Key stakeholders from each district collaborated with project staff to design district-specific MWH mission and vision statements, draft bylaws for MWH governance committees and set parameters around terms and composition of governance committees. On average, governance committees had 9 individuals, ranging from 6 to 12 upon establishment [21]. Governance committee members were trained in principles of governance and their roles in ensuring adherence to formalized MWH policies and overseeing long-term finances and operations [16]. Governance committee members received monthly mentorship visits from project staff and attended mid-project workshops to review collective lessons learned.

*Management units.* Governance committees then selected management units of 1 or more individuals to oversee daily operations of the MWHs, operating on a rotational basis. On average, management units had 5 individuals, ranging from 1 to 10 [21]. Management units were tasked with maintaining MWH registers and assets, managing daily operations, and conducting daily cleaning. Management units received monthly mentorship visits from project staff concurrent with the governance committees.

**MWH stakeholder engagement.** To foster buy-in and ownership, ongoing, targeted stakeholder engagement activities were conducted throughout the project. MAHMAZ engaged stakeholders in government at the national, provincial, district, and local levels; with traditional leadership at the chiefs and village headmen levels; with the network of organizations operating in maternal child health; and with local communities surrounding the MWH sites. A project launch was held at the national level; community launches were held at each site upon opening of the MWHs. Each launch was preceded by a series of preparatory meetings with relevant stakeholders at that level. Additionally, project staff hosted meetings with each governance committee and the local headmen to foster learning and accountability of the governance committee to the local populations, and to solicit additional community contributions for the MWHs (donations of money, maize, livestock, etc.) [21]. While smaller meetings also occurred with various stakeholders, as well as submission of frequent letters and reports, neither resulted in meaningful project costs. The stakeholder engagement activities relevant for this costing description included the project launch, site launches, and community meetings. A summary of these activities is provided in S1 Table.

**Study design and analysis.** We conducted a retrospective, top-down, program cost description (partial economic evaluation) using project document review to identify planned and actual direct financial costs of setting up a MWH [7]. In accordance with the principles of top-down costing, we used overall, or top level, expenditures (by category where appropriate), to estimate service costs (in this case, cost per visit to a MWH) [22]. Planned and actual costs are presented and highlight variance between our initial budget estimations and actual expenditures. Standard costing methods were used to estimate the annual equivalent cost per visit to the MWH [23]. We also ensured that our methods aligned with the principles described in the Global Health Cost Consortium Reference Case for Estimating the Costs of Global Health Service and Interventions checklist (GHCC Reference Case). This tool outlines best practices for costing for policy makers, program managers, research institutions, non-governmental organizations, and other bodies [24].

This analysis focuses exclusively on the MWH financial setup costs. We define setup costs as the full profile of expenditures to complete initial construction and furnishing the MWHs, and to implement the initial capacity building and other activities that were conducted to engage key stakeholders and benefactors in the MWH setup. These setup activities differ from the subsequent operating activities, and thus present a unique cost profile. Costs related to

operating and sustaining the MWHs were planned to be covered by income-generating activities which were implemented at each MWH. These costs are not included in this analysis, but will be described elsewhere.

**Planned and actual project costs.**   At time of procurement, all purchase and expense receipts were scanned, saved on a secure server, and documented in a tracking database for financial accountability and charging to the grants. We reviewed all expenses charged to the grants from July 2015 through January 2019, the total length of the project. Using that tracking database to ensure all receipts were reviewed, expense data were retrospectively extracted on encrypted tablets using SurveyCTO Collect Software (Dobility, Inc. 2021). We extracted the voucher number (internal unique identifier for the financial document), expense date, and expense description. Using a pre-defined coding system and using the expense description and dates, all expenses were assigned a cost code by the individual extracting the data to group the expenses by activity type and expense purpose. In a few instances where the relevant project activity was not clearly included on the expense documents, it was imputed based on the description provided, date of expense in relation to the project workplan, and position of the person who incurred the expense. Expense data were cleaned using SAS v9.4 software (SAS Institute Inc., Cary, NC, USA). All expenses were recorded in Zambian Kwacha (ZMW) at the time of procurement. During analysis, costs were inflated to reflect 2022 costs. Finally, they were adjusted to US Dollars (USD), using the average exchange rate from the first 6-months of 2022 [24, 25].

Lower-level costing codes were grouped into two overarching categories and four subcategories: "capital" costs relating to the physical MWH building and tangible furnishings, and those classified as "installation" costs, relating to one-time, non-recurring activities and trainings during setup period. Capital costs included subcategories: infrastructure and furnishing; installation costs included subcategories: capacity building and stakeholder engagement.

Planned costs were extracted from the primary budget initially submitted to the funder. The field-based costs within the budget were retrospectively grouped into the 4 subcategories mentioned above using line-item cost descriptions for expenses; examples of such descriptions included: contractor service agreements, construction insurance, meeting venue/food, per diem, transport, printing, etc.

Planned and actual costs were summed and compared in Microsoft Excel, organized by the 4 sub-categories. We present costs as an average per site. In many instances costs were recorded at the program level and could therefore not be attributed to specific sites. Such costs were allocated across sites the 10 sites. For example, the primary labor contract was presented as one flat fee, not broken up by site; This cost was then allocated across the 10 sites to get a site-level cost. Variances between planned and actual costs per site are presented as percentages. Additionally, we conducted a scenario analysis to consider how shifts in costs–namely infrastructure costs–may impact the total program cost. We did this by inflating the infrastructure costs by 25% and 50% to mimic medium or high construction cost environments.

**Estimating annual costs per site.**   All costs occur upfront but retain value over their useful life. We annuitized the costs over their useful life using a real discount rate of 3% per annum as there is no specified rate for economic evaluations in Zambia [23, 24, 26, 27]. The expected useful life of the cost sub-categories was based on the literature and standard practice. The MWHs were constructed from cement blocks and are estimated to have a lifespan of 30 years [28]. Other furnishings including beds, mattresses, etc. were estimated at 5 years [29, 30]. Project trainings, supervision, and capacity building were estimated at 3 years, given expected turnover given the term limits of the governance committee [21]. The equivalent annual cost was estimated by dividing the total cost by the relevant annuity factor.

**Cost per MWH visit.**   We used data routinely captured by the project to calculate MWH utilization. On a monthly basis, individual level data were extracted monthly from MWH

registers by project data collectors, which allowed us to calculate number of MWH visits and bed-nights for the study period [31]. Utilization data were restricted to the final 12 months of our study period (August 2017 –July 2018) which is generally reflective of standard operations of the MWHs. To assess utilization, we calculated overall occupancy by dividing the number of bed-nights (date of discharge minus the day of visit) per month by the total available bed-nights (number of beds multiplied by the number of nights per month). We stratified by the space allocated for those awaiting delivery (10 beds) and the space for those who traveled to attend a postnatal visit (4 beds). In cases where date of visit or discharge was missing, we used single imputation based on visit type to estimate bed-nights. We then calculated annuitized costs per visit and cost per bed-night. Cost per visit and per bed-night are calculated only for startup costs, not operating costs. We display these costs to highlight how this component of future costs may change with different levels of utilization.

## Sensitivity and scenario analysis

To account for uncertainty, we also conducted sensitivity and scenario analyses to hypothesize about alternative scenarios. We first conducted a univariate sensitivity analysis to model varying discount rates. In addition to annuitizing costs using a 3% discount rate, we also calculated costs using 2% and 5% discount rates.

Cost per visit and cost per bed-night are dependent on utilization and so we present 2 alternative theoretical scenarios, low occupancy (25%) and high occupancy (85%), to explore how cost per visit varies from the observed. We present average site occupancy of the 10 sites, but the theoretical scenarios convey how cost per visit may differ based on site-specific occupancy rates. To estimate the number of visits for the theoretical scenarios, we assumed the same ratio of visits to bed-nights (or average length of stay) that we observed in our data. We stratify the overall cost into capital and installation components.

**Excluded costs.** Only actual and planned direct costs associated with the setup of the MWHs were included in this analysis. Research expenses related specifically to the impact and process evaluations, project overhead (office space, office supplies, vehicles, etc.), and staff salaries were excluded. All recurrent operating costs for the MWHs were excluded from this analysis as these were assumed to be covered by each MWH's income generating activities (to be reported elsewhere). The costs included are considered indicative of the cost of replicating the MWH setup outside of a study setting. We also do not include long run operations costs likely to be incurred after the initial setup period is complete.

**Ethics.** Ethical approval for the MAHMAZ implementation evaluation was granted by the Boston University Institutional Review Board (IRB) (protocol H-35321) and the ERES Converge IRB in Zambia (reference number 2016-June-023) [31]. Additional permissions were granted by the National Health Research Authority, the Ministry of Health (MOH), and traditional leaders. The costing data is not considered human subjects data. The MWH register was considered by both IRBs to be routine programmatic data. Consent was waived for extracting the data from the MWH register on the grounds that it was not feasible or practicable, and all data would be presented deidentified and aggregated. Written informed consent was obtained for all interviews and surveys in the larger implementation study, though those data are not presented in this manuscript.

## Results

The average cost to set up one MWH, inclusive of infrastructure, furnishings, capacity building and stakeholder engagement activities, was US$85,284, 6% lower than the budgeted cost (Table 2). The infrastructure costs made up the majority (62%) of the actual project costs but

**Table 2. Planned and actual direct costs incurred by MAHMAZ project to set up 10 new MWHs by cost category, presented in USD.**

| Cost Categories | Planned Cost per Site[†] | *Actual* Cost per Site[†] | Variance | Estimated Lifespan | Annuitized Cost[†] ** |
|---|---|---|---|---|---|
| **Capital Costs** | | | | | |
| **Infrastructure** | $55,788 | $53,240 | 5% | 30 years | $2,716 |
| *MWH structure* | (61%) | (62%) | | | (22%) |
| **Furnishing** | $15,812 | $11,311 | 28% | 5 years | $2,470 |
| *Beds, bedding, cooking supplies, etc.* | (17%) | (13%) | | | (20%) |
| **Subtotal** | **$71,600 (79%)** | **$64,551 (76%)** | **10%** | | **$5,186 (41%)** |
| **Installation\* Costs** | | | | | |
| **Capacity Building** | $15,763 | $12,336 | 22% | 3 years | $4,361 |
| *Trainings, mentorship* | (17%) | (14%) | | | (35%) |
| **Stakeholder Engagement** | $3,391 | $8,397 | (148%) | 3 years | $2,969 |
| *Site launches, leadership meetings* | (4%) | (10%) | | | (24%) |
| **Subtotal** | **$19,154 (21%)** | **$20,773 (24%)** | **(8%)** | | **$7,330 (59%)** |
| **Grand Total** | **$90,754 (100%)** | **$85,284 (100%)** | **6%** | | **$12,516 (100%)** |

*Installation costs for the purpose of this paper include trainings, mentorship, and stakeholder engagement relevant to the establishment of the intervention. Subsequent activities, and day to day running costs are excluded from this analysis.

**Annuitized costs were calculated using a 3% discount rate using the following annuitization factors: 19.6004 for 30-year lifespan; 4.5797 for 5-year lifespan; 2.8286 for 3-year lifespan [32]

†Figures in parentheses indicate percentages of the total costs

comprised only 22% of costs when annuitized. Installation costs comprised 59% of total annuitized costs. Variance between planned and actual costs was highest in the stakeholder engagement sub-category, where the actual costs were more than double the planned costs. While the stakeholder engagement is 10% of actual total cost per site, it comprises 24% of annuitized total cost.

Each MWH was roughly 250 square meters (including a 70 square meter, concrete-walled, open-air drying area and a separate covered cooking space) and was built at an estimated cost of space of USD$213 per square meter (USD$53,240 divided by 250). Excluding the open-air drying space, which had neither concrete foundations nor roofs, each home was 180 square meters and cost USD$296. Over three quarters of the capital costs were expended on the construction contract while approximately two thirds of the installation costs spent on meals, incidentals and accommodation expenses for staff and training/meeting participants (Fig 1).

Ten of the 14 beds (71.4%) in each MWH were designated for women awaiting delivery; the remaining 4 beds (28.6%) were designated for women awaiting PNC. Average length of stay was 11.2 bed-nights for all visit types but varied by type of stay: Women stayed an average of 13.4 nights while awaiting delivery, and 2.5 nights while staying for PNC. MWH occupancy varied by site, however the overall occupancy rate was 39% between August 2017 and July 2018. The occupancy rate for delivery beds was 52%, while PNC-only allocated space was 6%. The average overall cost per bed-night was $6; $5 per night for those awaiting delivery and $41 per night for those seeking postnatal care (Table 3).

The costs per visit overall and by capital and installation only vary by volume with generally 50% lower costs in the theoretical peak occupancy scenario (85%) and more than 50% higher costs in the low occupancy scenario (25%). When annuitized, capital-only costs (infrastructure and furnishings) per stay were only about $3 per bed-night ($29 per visit).

Total program cost per annum was $12,516 at a 3% discount rate. In conducting our sensitivity analysis, we found that total program cost per annum was $11,966 at a 2% discount rate, $13,689 at a 5% discount rate, and $16,968 at a 10% discount rate (S3 Table). This yielded a

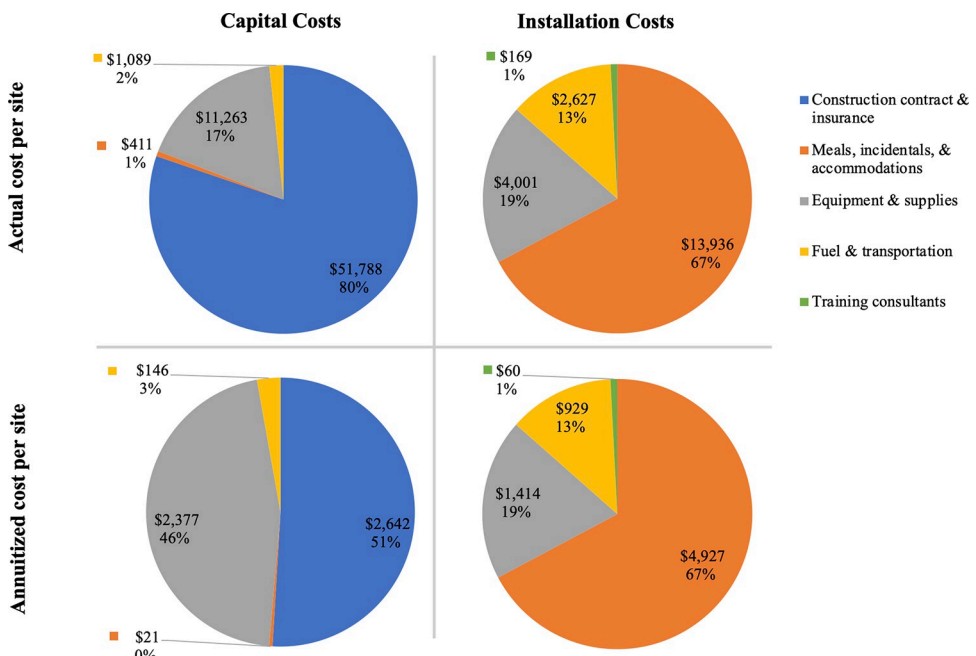

**Fig 1. Per site detailed distribution of cost categories across capital and installation categories, by total actual costs and annuitized costs.**

range of $5,002 between the 2% and 10% rates. When considering a scenario analysis by inflating the infrastructure costs by 25%, infrastructure costs were $66,550, and total project cost was $98,594 (16% relative change). When inflating the same costs by 50%, infrastructure costs were $79,860 and total project cost was $111,905 (31% relative change) (S4 Table).

## Discussion

The MAHMAZ project built 10 MWHs and implemented capacity building governance activities and stakeholder engagement at each MWH in rural Zambia. Predictably, a large proportion of actual setup costs were related to the capital costs including constructing and furnishing the MWHs. While costs may be lower using different materials or by sourcing materials from local communities (i.e., burnt bricks), we deliberately sought a professional contractor who proposed high quality materials in order to maximize the lifespan of the building. While construction materials were of high quality, decorative finishes were not added to the construction to save on cost. Despite being constructed in rural areas which necessitated transportation costs, the estimated construction costs of USD$296 per square meter are lower than other published estimates of $400-$1,000 per square meter for buildings of similar material and quality in Zambia and the surrounding region [33, 34].

Stakeholder engagement is a critical aspect of program implementation, facilitating local buy-in at multiple levels, utilization of the intervention by target populations, and long-term sustainability [35]. Important installation start-up costs including stakeholder engagement activities and capacity building for the governance committees comprised only 24% of actual costs incurred, but 59% of the annuitized costs per site due to the short predicted lifespan of the activities of only 3 years. When budgeting, we underestimated the cost of stakeholder engagement by half. In reality, the project invested heavily in site launches at each MWH and providing per diems for participants and staff during planning meetings and workshops. Site

**Table 3. Annuitized cost per maternity waiting home stay over a 12-month period, by bed space allocation, occupancy rates (actual vs. theoretical low and high scenarios), and type of costs (total, capital, and installation).**

| | All visit types (all 14 beds) | | | Delivery visits only (10 beds) | | | PNC visits only (4 beds) | | |
|---|---|---|---|---|---|---|---|---|---|
| | Actual use | Theoretical use | | Actual use | Theoretical use | | Acutal use | Theoretical use | |
| **Occupancy rate** | **39%** | **25%** | **85%** | **52%** | **25%** | **85%** | **6%** | **25%** | **85%** |
| **Total MWH visits, n[1]** | 178 | 114 | 388 | 142 | 68 | 232 | 35 | 146 | 496 |
| **Total MWH bed nights, n[1]** | 1,989 | 1,278 | 4,344 | 1,901 | 913 | 3,103 | 88 | 365 | 1,241 |
| **MWH project total costs, annuitized (USD)[2]** | $12,516 | $12,516 | $12,516 | $8,940 | $8,940 | $8,940 | $3,576 | $3,576 | $3,576 |
| Cost per MWH visit | $70 | $110 | $32 | $63 | $131 | $39 | $102 | $24 | $7 |
| Cost per MWH bed-night | $6 | $10 | $3 | $5 | $10 | $3 | $41 | $10 | $3 |
| **Capital costs, annuitized (USD)** | $5,186 | $5,186 | $5,186 | $3,704 | $3,704 | $3,704 | $1,482 | $1,482 | $1,482 |
| Cost per MWH visit | $29 | $45 | $13 | $26 | $54 | $16 | $42 | $10 | $3 |
| Cost per MWH bed-night | $3 | $4 | $1 | $2 | $4 | $1 | $17 | $4 | $1 |
| **Implementation costs, annuitized (USD)** | $7,330 | $7,330 | $7,330 | $5,236 | $5,236 | $5,236 | $2,094 | $2,094 | $2,094 |
| Cost per MWH visit | $41 | $64 | $19 | $37 | $77 | $23 | $60 | $14 | $4 |
| Cost per MWH bed-night | $4 | $6 | $2 | $3 | $6 | $2 | $24 | $6 | $2 |

MWH = maternity waiting home

[1] Actual visits and nights stayed from the Maternity Waiting Home register

[2] Project budget (annuitized using a 3% discount rate) multiplied by percent of space allocated. Total beds = 100% of space; Delivery beds (10) = 71.4% of space; PNC beds (4) = 28.6% of space

• All visit types = 5,110 possible bed-nights; Delivery = 3,650 possible bed-nights; PNC = 1,460 possible bed-nights

• Actual use reflects visits to the MWH observed during the final 12 months of evaluation. Actual occupancy was calculated using visits divided by the total possible bed nights = All visit types = 5,110 possible bed-nights; Delivery = 3,650 possible bed-nights; PNC = 1,460 possible bed-nights

• Theoretical use estimates 25% and 85% occupancy based on total possible bed nights (described above)

• Costs are based on startup costs and exclude operating costs.

launches and headmen meetings were not initially planned, but it became apparent during the iterative process of community engagement that they would be a critical piece to ensure community buy-in and utilization of MWHs. Site launches, in particular, were requested by multiple levels of government and by external funders. While they are only a small proportion of overall cost, the activities and resultant support proved critical to project success: engagement at the district and health facility levels encouraged health system ownership over the MWHs; community engagement encouraged utilization of the MWHs, a sense of community ownership over the program, and a desire to hold governance committees accountable to the community at large [21, 36]. The donors' flexibility was instrumental, as it allowed us to re-budget and allocate additional funds to these activities; other funders may not allow such re-budgeting. The activities to build the capacity of the governance committees were important to ensure smooth operations of the MWH both during and post-project: community members felt involved and responsible for the upkeep of the MWHs and felt driven to protect the future of the MWH [36].

While we set up the project over 42 months for planning, construction, training and mentorship of governance committees, and stakeholder engagement, in fact, to promote full sustainability, support should be extended through several years of operation. Hindsight would have us continue mentoring the governance committees through more than one year of operations after opening the MWHs to oversee multiple cycles of re-elections and trainings of new committee members; attend several open public meetings where governance committees report activities and finances to foster routine accountability to the local community [21]; support and increase ownership over data collection and analysis of feedback from women; and

further assist with day-to-day operations including hiring and firing, among others. This would have further instilled confidence in the governance and management systems, and increased sustainability and long-term success. It would also have increased the setup costs. Other implementers may require less time in the planning phase with the resources provided here and elsewhere in the literature [7, 21].

In response to community input gathered during the intervention design phase, MAHMAZ built each MWH to have a separate small space allocated for women awaiting PNC visits, as it was not culturally acceptable for newborns to reside in the same space with pregnant women [15]. However, PNC visits to the MWH overall and per bed-night were costly due to low utilization by women awaiting a PNC. Despite the comparatively low utilization and high costs, MWH were found to improve PNC utilization among women living most remotely in study intervention sites compared to control sites [37]. Balancing community needs and programmatic costs is challenging, particularly in such rural settings where utilization is low. Other implementers should weigh the costs and benefits of including PNC space within the MWH when utilization is likely to be low.

In this intervention, both capital (infrastructure and furnishing) and installation (capacity building and stakeholder engagement activities) were necessary. However, while we present the cost of each component separately, we are unable to assess what the utilization would look like if we only implemented one component or the other. Some implementers may have interest the sustainability of new MWHs in the absence of governance systems and stakeholder engagement. In contrast, some communities that have existing shelters may consider the utility of establishing governance committees alone [38]. A three armed cluster control trial in Ethiopia evaluated 1) improving MWHs and local leader training, and 2) local leader training alone without any MWH improvements, and 3) the standard of care. Results from this trial found that both intervention arms displayed small non-significant increases in facility delivery compared to the standard of care [38]. Further, it provides evidence that some implementers may want to utilize local leader training–in our work, installation costs–but not the entire MWH construction. For this reason, we presented costs separately, anticipating that some governments or implementing organizations may find value in planning or implementing each component individually.

A recently published review detailed the dearth of economic analyses for implementation science research. There is very little published knowledge about implementation cost descriptions and outcomes [39]. Prior to this study, there has been little research or guidance for implementers on the initial investment costs needed to set up MWHs, or the costs per outcome. Most studies analyzing costs focused on user expenditures [40] or willingness of users to pay [41, 42]. There are a number of systematic reviews and other studies that aimed at comparing costs to strengthen maternal and child health systems in LMIC, but the studies are either not comparable or have not yet published results [43–46]. In addition, few report start up programmatic costs [47, 48].

Given our limited ability to compare installation costs with other interventions aiming to improve facility delivery and ultimately reduce maternal mortality, it is beneficial to anchor our results within the context of the Zambian MOH budget. In 2019, approximately USD $8,114,966 was allocated toward construction of three district hospitals and 108 health posts. The costs for constructing and furnishing a MWH in our program, was $64,551, or 0.8% of the 2019 MOH infrastructure budget. Additionally, in 2019, 0.6% of the MOH budget was allocated to individual District Health Service Delivery accounts: districts received an average of USD$323,352 (range $97,604 - $690,765). In the districts where our program was implemented, setting up one MWH would comprise 12–31% of the District Health Service Delivery budgets, depending on the district. Capital costs per MWH would comprise 9–24% of the

Health Service Delivery budgets for the study districts; Capacity building and stakeholder engagement costs per MWH would comprise 3–8% of the same budget, though there would likely be spillover and efficiencies to be gained with capacity building efforts.

Though a decision to invest in MWHs depends on many factors, findings from the MHA evaluation suggest that the Core MWH Model implemented here significantly increased facility delivery and access to higher level care when needed among women living greater than 10 km from their designated rural health center, directly addressing this distance barrier, a commonly cited challenge to accessing maternity care in rural areas [37, 49, 50]. The key findings of the impact evaluation also include improvements in MWH utilization, exposure to maternal and well-baby counselling, and PNC attendance, suggesting the benefits of this MWH model extend along the care continuum [51]. One key challenge for implementers and decision makers to consider, of course, is that while MWHs may be most beneficial for women living remotely, the costs per visit and per bed-night are directly dependent on volume, and remote areas are generally not densely populated.

## Limitations

This study has several limitations. First, while ascertaining the actual costs was straightforward, the planned costs were difficult to unpack because the budgets were revised and adapted as the project evolved and additional funding secured. Early budgets used for this analysis may have incorrectly estimated costs of components, while subsequent iterations became informed by actual expenditure. To best represent initial planning, we only used the first budgets submitted to the funders under the grants to estimate planned costs. Second, this analysis presents cost per visit and per bed-night. These indicators are on the theoretical pathway to a health outcome, but this analysis does not estimate the cost per health outcome or cost effectiveness. Third, we were unable to estimate the influence of location on MWH costs; presumably transportation of construction equipment and materials to more remote settings would have been costlier, or larger catchment areas would require more stakeholder engagement efforts. Lastly, as literature is scarce, it was challenging to compare the setup costs of this intervention to others with similar aims. However, providing this information for a MWH intervention adds to the sparse literature and will serve as a tool to guide implementers considering similar interventions.

## Conclusion

The World Health Organization recommends MWHs as a strategy to reduce maternal mortality rates. This analysis serves as a planning resource for governments and implementing partners that are considering MWHs as a component of their overall maternal and child health strategy.

## Supporting information

**S1 Questionnaire. Inclusivity in global research questionnaire.**
(DOCX)

**S1 Table. Overview of project-facilitated activities.**
(DOCX)

**S2 Table. Main structural elements and furnishing for the MWHs provided by the MAH-MAZ project.**
(DOCX)

**S3 Table. Univariate sensitivity analysis of annuitized financial costs incurred by MAH-MAZ project, using variable discount rates.**
(DOCX)

**S4 Table. Theoretical costs assuming variable inflation of infrastructure costs.**
(DOCX)

**S1 Fig. Architectural drawings used to construct MAHMAZ MWHs.**
(TIF)

**S2 Fig. Photographs of MAHMAZ MWHs.** Components from left to right, top to bottom: a. New and old MWH structures; Credit: Jeanette Kaiser, Boston University School of Public Health, 2017. b. New MWH structure; Credit: Jeanette Kaiser, Boston University School of Public Health, 2017. c. Main dormitory with 10 beds, bedding, mosquito nets, and cabinets; Credit: Wakunyambo Imasiku, Zambia Center for Applied Health Research and Development, 2017. d. Cooking shelter; Credit: Oliver Malupande, Zambia Center for Applied Health Research and Development, 2017. e. Courtyard and drying racks; Credit: Jeanette Kaiser, Boston University School of Public Health, 2017. f. Office and storage room with a desk, cooking pots, cups, and bedding; Credit: Jeanette Kaiser, Boston University School of Public Health, 2017. g. Latrines; Credit: Jeanette Kaiser, Boston University School of Public Health, 2017. h. Verandah; Credit: Wakunyambo Imasiku, Zambia Center for Applied Health Research and Development, 2017.
(TIF)

## Acknowledgments

The authors would like to thank the Zambian Ministry of Health at the National, Provincial, and District levels, as well as the traditional leadership of the relevant areas, for their approval and support. We appreciate the assistance of health facility staff, and the maternity waiting home governance committee and management unit members for their contributions to the program. We would also like to thank the following individuals who played an integral role in project management and implementation: Denson Chongwe, Parker S Chastain, and Carey Howard. We appreciate Jason Usher and Nicollette Johnson who contributed to entry of the costing data into our electronic data capture system, and Kathleen L McGlasson who managed the incoming data.

## Author Contributions

**Conceptualization:** Allison Juntunen, Nancy A. Scott, Jeanette L. Kaiser, Taryn Vian, Viviane Sakanga, David Kalaba, Lawrence C. Long.

**Formal analysis:** Allison Juntunen, Misheck Bwalya, Lawrence C. Long.

**Funding acquisition:** Nancy A. Scott.

**Investigation:** Allison Juntunen, Nancy A. Scott, Jeanette L. Kaiser, Taryn Vian, Thandiwe Ngoma, Kaluba K. Mataka, Viviane Sakanga, David Kalaba, Godfrey Biemba, Peter C. Rockers, Davidson H. Hamer, Lawrence C. Long.

**Methodology:** Nancy A. Scott, Jeanette L. Kaiser, Lawrence C. Long.

**Project administration:** Nancy A. Scott, Thandiwe Ngoma, Kaluba K. Mataka, Misheck Bwalya, Viviane Sakanga.

**Supervision:** Nancy A. Scott, Jeanette L. Kaiser, Taryn Vian, Lawrence C. Long.

**Writing – original draft:** Allison Juntunen, Nancy A. Scott, Jeanette L. Kaiser, Lawrence C. Long.

**Writing – review & editing:** Allison Juntunen, Nancy A. Scott, Jeanette L. Kaiser, Taryn Vian, Thandiwe Ngoma, Kaluba K. Mataka, Misheck Bwalya, Viviane Sakanga, David Kalaba, Godfrey Biemba, Peter C. Rockers, Davidson H. Hamer, Lawrence C. Long.

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
