## [Decision Letter · Decision Letter 0]

21 Jul 2022

PGPH-D-22-00186

An economic evaluation of the setup costs of community-owned maternity waiting homes in rural Zambia

Dear Dr. Juntunen,

Thank you for submitting your manuscript to PLOS Global Public Health. After careful consideration, we feel that it has merit but does not fully meet PLOS Global Public Health’s publication criteria as it currently stands. Therefore, we invite you to submit a revised version of the manuscript that addresses the points raised during the review process.

We look forward to receiving your revised manuscript.

Kind regards,

Sarker AR, PhD

Academic Editor

Journal Requirements:

2. Please ensure that the funders and grant numbers match between the Financial Disclosure field and the Funding Information tab in your submission form. Note that the funders must be provided in the same order in both places as well.

3. Please update your online Competing Interests statement. If you have no competing interests to declare, please state: “The authors have declared that no competing interests exist.”

4. In the online submission form, you indicated that your data will be submitted to a repository upon acceptance.  We strongly recommend all authors deposit their data before acceptance, as the process can be lengthy and hold up publication timelines. Please note that, though access restrictions are acceptable now, your entire data will need to be made freely accessible if your manuscript is accepted for publication. This policy applies to all data except where public deposition would breach compliance with the protocol approved by your research ethics board. If you are unable to adhere to our open data policy, please kindly revise your statement to explain your reasoning and we will seek the editor's input on an exemption. Please be assured that, once you have provided your new statement, the assessment of your exemption will not hold up the peer review process.

5. Please provide separate figure files in .tif or .eps format and ensure that all files are under our size limit of 10MB.

6. We noticed that you used “unpublished data” in the manuscript. We do not allow these references, as the PLOS data access policy requires that all data be either published with the manuscript or made available in a publicly accessible database. Please amend the supplementary material to include the referenced data or remove the references.

7. We do not publish any copyright or trademark symbols that usually accompany proprietary names, eg (R), (C), or TM  (e.g. next to drug or reagent names). Please remove all instances of trademark/copyright symbols throughout the text, including ® (SurveyCTO®, Microsoft®) on pages 12 and 13.

8. All figures and supporting information files will be published under the Creative Commons Attribution License (creativecommons.org/licenses/by/4.0/). Authors retain ownership of the copyright for their article and are responsible for third-party content used in the article. 

S1 Figure: Please confirm (a) that you created the image/clip-art in the figure panels; or (b) replace with an open source alternative. See these open source resources you may use to replace images/clip-art:

- http://www.publicdomainpictures.net/

- https://openclipart.org/

S2 Figure: Please confirm (a) that you are the photographer; or (b) provide written permission from the photographer to publish the photo(s) under our CC-BY 4.0 license.

Please upload any written confirmation as an 'Other' file type. It must clarify that the copyright holder understands and agrees to the terms of the CC BY 4.0 license; general permission forms that do not specify permission to publish under the CC BY 4.0 will not be accepted. Note that uploading an email confirmation is acceptable.

Additional Editor Comments (if provided):

The title is misleading and should be revised based on the objective of this study. Please follow book written by Michael F. Drummond for costing study and revise the paper accordingly.

Reviewers' comments:

Reviewer's Responses to Questions

**Comments to the Author**

1. Does this manuscript meet PLOS Global Public Health’s publication criteria? Is the manuscript technically sound, and do the data support the conclusions? The manuscript must describe methodologically and ethically rigorous research with conclusions that are appropriately drawn based on the data presented.

Reviewer #1: No

Reviewer #2: Yes

2. Has the statistical analysis been performed appropriately and rigorously?

Reviewer #1: No

Reviewer #2: Yes

3. Have the authors made all data underlying the findings in their manuscript fully available (please refer to the Data Availability Statement at the start of the manuscript PDF file)?

Reviewer #1: No

Reviewer #2: Yes

4. Is the manuscript presented in an intelligible fashion and written in standard English?

Reviewer #1: Yes

Reviewer #2: Yes

5. Review Comments to the Author

Reviewer #1: In this study the authors described the set-up costs of ten Maternity Waiting Home in Zambia. Studies related to MWHs are very relevant especially in settings with a low access to delivery and postnatal care services. The authors raised an important public health issue for low income country.

However, the study has lots of shortcomings:

In the title the authors indicated that they did an economic evaluation which was not revealed in the objective and rest of the paper. The authors may refer books and articles to understand and conduct an economic evaluation. There are full and partial economic evaluation. The book written by Michael F. Drummond and his colleagues is a good book for anyone interested to conduct an economic evaluation of healthcare programs. You may follow the link: https://books.google.com.et/books/about/Methods_for_the_Economic_Evaluation_of_H.html?id=yzZSCwAAQBAJ&redir_esc=y

A good article by Luke Rudmik, and Michael Drummond, published in 2013 defines economic evaluation as: “…is the comparative analysis of alternative courses of action in terms of both their costs and consequences. With increasing health care expenditure and limited resources, it is important for physicians to consider the economic impact of their interventions.”

https://onlinelibrary.wiley.com/doi/10.1002/lary.23943

However, there is nothing said about consequence in the study’s contents. Even the costs described in the study are simple reports of set-up costs. The core thing in research, which is a systematic search some issue to establish facts and reach new conclusions were not addressed sufficiently in the study. The authors simply collected some financial data and presented that in a table.

The authors may collect additional relevant data about service costs, users' costs and some measure of consequences to improve their study. It would be better if they take some comparator in their analysis, even do nothing could be a comparator.

Reviewer #2: I really like this paper. Its purpose is clear, and its methods are clearly described for the most part (see comments below). It is useful information for others maternal health program planners who may be interested in costs of constructing MWH, and how these evolve over time and based on utilization. I provide comments below for strengthening clarity of the paper.

Methods

1. Page 12, line 242, I’d recommend that the authors use the term ‘Start-up’ costs relating to one-time, non-recurring activities, rather than ‘implementation costs’. Implementation of an intervention can occur during a start-up and on-going phases. Start-up costs are a common term used in global health costing studies. If they want to refer to one-time, non-recurring activities as something other than ‘Start-up’, I’d suggest, ‘Installation’, rather than ‘Implementation’. Typical ‘Start-up’ cost categories are infrastructure and furnishing, capacity building (training), stakeholder engagement. Others may include materials development, for example. They could omit the foot note in table 2, if they used the terminology ‘Start-up’ costs or ‘Installation’ costs. IF changed, they would need to change headers in Figure 1 and in other places in the text.

2. Page 12, lines 252-253: These two sentences could use a little rewording. “We present costs as an average per site. In many instances costs were recorded at the program level and could therefore not be allocated to specific sites. Costs by category were therefore evenly allocated across sites by dividing total costs by the number of sites (10). “

a. “Costs as an average per site, or do you mean ‘average set up costs’—i.e. sum of total set up costs across all sites divided by number of sites. I think the authors derive an average across all sites. This is just an average cost for setting up the maternity waiting home.

b. They indicate that many costs couldn’t be allocated to specific sites, which is expected. Were ALL costs allocated across sites, or just some of the shared program costs? If the latter could they add a little descriptive, such as “For example, some shared program costs, such as x, y and z, were allocated evenly across each site by dividing the total shared program costs by the number of sites (10 sites). These costs were then added to the site-specific costs. (Or something to that effect)

3. Page 13, line 280-286—I’m not sure why the authors refer to cost per admission and cost per bed-night, since these costs do not include operating costs. Cost per bed-night would typically include operating costs.

4. I think it is fine to present a unit cost based on different scenarios of utilization, but just confused by reference to cost per admission and cost per bed-night. Can the authors present a clearer rationale for this, or is it possible to simply refer to ‘Cost per MWH visit’ under different occupancy scenarios and calculations?

5. The authors do not conduct a sensitivity analysis, and it would make the results more generalizable to other settings if they could do this. At a minimum, it would be useful to vary the discount rate of 3%, since for capital infrastructure investments, this assumption is critical. If there are any other uncertain variable estimates, they could vary those as well. I wonder if they could consider international vs local prices for key construction materials, or consider varying labor costs, if Zambia construction labor is much higher or lower than other Southern African countries. I don’t think it would be a heavy lift to do univariate sensitivity analysis for this paper.

a. The authors note in the discussion that the construction s costs were significantly lower than other published estimates for Zambia, so sensitivity analysis may help to strengthen robustness of the authors results.

6. Minor: Indicate somewhere in the paper, you are analyzing Financial costs. (i.e. you do not include economic or opportunity costs in this analysis). Also indicate what year USD are in, i.e. 2020 USD or? All tables and figures should clearly indicate the USD (year) in the table or figure title.

Results

1. Page 14, line 311. Should this read, “The average cost to set up one MWH….”

a. Do the authors have disaggregated costs by site, so they could include that in an appendix table? This would show the site-specific costs for each site, and then the allocated shared program costs. This is related somewhat to the above question on how average costs were estimated.

b. Lines 313 to 315 provide cost shares, however neither Table 2 or Figure 1 show this. This sentence is confusing: “The infrastructure costs made up the majority (62%) of the actual project costs but comprised only 21% of actual costs when annuitized.”

i. I think it may read better as: The infrastructure costs made up the majority (62%) of the actual project costs but comprised only 21% of costs when annuitized.” (DELETE ‘ACTUAL’)

c. I’d recommend that in Table 2, the authors add (cost shares, % of total cost) in parenthesis in each column where cost data are presented: Planned cost per site, Actual Cost per site and Annuitized cost. Then the reader could see this more clearly.

Discussion

1. Page 20, lines 411-414, this is confusing. Did the authors present their costs to a study in Ethiopia? I may suggest deleting lines 411 to 414 or elaborating a bit more.

2. Kudos to the authors for anchoring the results within the context of the Zambian MOH budget. Nicely done!

3. Lines 415 to 446 could likely be synthesized a bit more and shortened, i.e. there are number of systematic reviews aimed at comparing costs to strengthen maternal and child health systems in LMIC, but the studies are either not comparable nor have they published their results. In addition, few report start up programmatic costs (43, 44, 45, and ?)

a. Minor: Line 440, reference for systematic review?

Suggestion: The authors may want to check their analysis against the Global Health Cost Consortium Reference Case check list. I think they adhere to it, with one or two exceptions. Available on line https://ghcosting.org/pages/standards/reference_case

6. PLOS authors have the option to publish the peer review history of their article (what does this mean?). If published, this will include your full peer review and any attached files.

**Do you want your identity to be public for this peer review?** For information about this choice, including consent withdrawal, please see our Privacy Policy.

Reviewer #1: No

Reviewer #2: No

---

## [Editor Report · Decision Letter 1]

29 Nov 2022

PGPH-D-22-00186R1

A cost description of the setup costs of community-owned maternity waiting homes in rural Zambia

Dear Dr. Juntunen,

Thank you for submitting your manuscript to PLOS Global Public Health. After careful consideration, we feel that it has merit but does not fully meet PLOS Global Public Health’s publication criteria as it currently stands. Therefore, we invite you to submit a revised version of the manuscript that addresses the points raised during the review process.

We look forward to receiving your revised manuscript.

Kind regards,

Sarker |AR, PhD

Academic Editor

Journal Requirements:

Additional Editor Comments (if provided):

Thanks for the revised version. I did not find any responses regarding the reviewers raised in original version of the manuscript. Please respond accordingly.
---

## [Decision Letter · Decision Letter 2]

23 Jan 2023

PGPH-D-22-00186R2

A cost description of the setup costs of community-owned maternity waiting homes in rural Zambia

Dear Dr. Juntunen,

Thank you for submitting your manuscript to PLOS Global Public Health. After careful consideration, we feel that it has merit but does not fully meet PLOS Global Public Health’s publication criteria as it currently stands. Therefore, we invite you to submit a revised version of the manuscript that addresses the points raised during the review process.

We look forward to receiving your revised manuscript.

Kind regards,

Abdur Razzaque Sarker, PhD

Academic Editor

Journal Requirements:

Additional Editor Comments (if provided):

Reviewers' comments:

Reviewer's Responses to Questions

**Comments to the Author**

1. If the authors have adequately addressed your comments raised in a previous round of review and you feel that this manuscript is now acceptable for publication, you may indicate that here to bypass the “Comments to the Author” section, enter your conflict of interest statement in the “Confidential to Editor” section, and submit your "Accept" recommendation.

Reviewer #2: (No Response)

Reviewer #3: (No Response)

2. Does this manuscript meet PLOS Global Public Health’s publication criteria? Is the manuscript technically sound, and do the data support the conclusions? The manuscript must describe methodologically and ethically rigorous research with conclusions that are appropriately drawn based on the data presented.

Reviewer #2: Yes

Reviewer #3: Partly

3. Has the statistical analysis been performed appropriately and rigorously?

Reviewer #2: Yes

Reviewer #3: Yes

4. Have the authors made all data underlying the findings in their manuscript fully available (please refer to the Data Availability Statement at the start of the manuscript PDF file)?

Reviewer #2: Yes

Reviewer #3: Yes

5. Is the manuscript presented in an intelligible fashion and written in standard English?

Reviewer #2: Yes

Reviewer #3: No

6. Review Comments to the Author

Reviewer #2: Thanks to the authors for their revisions. A few additional comments on the revised paper.

1. The first has to do with the sensitivity and scenario analysis. Ideally the authors would put their scenario and sensitivity analysis in one sub-section, describing them each to help understand the effect of these on the results, and the robustness of their results. In the methods section, they would move the description of changes in discount rate (line 279-281) and changes in infrastructure costs (lines 266-268) to the paragraph where they also vary assumptions on occupancy (lines 298 to 305). The sub-header could read Sensitivity and Scenario analysis.

a. The authors conduct a univariate sensitivity analysis of the discount rate, varying it to 2% and 5%. 3% is the low end, and I would recommend redoing that at a more realistic rate of 5% and 10%.

b. Then in results section, they would present their base case first, followed by estimates from the scenario and sensitivity analysis. I’d recommend moving lines 340 to 345 to around lines 365, when they also present the variation in occupancy.

2. Table 2 headers. Need to indicate that results presented in USD. Rather than add (%of total cost) in the column header, may want to put a footnote indicate that the numbers in parenthesis are % of total cost), otherwise the table is a little confusing as edited.

Reviewer #3: ABSTRACT:

#MAHMAZ in line 26 should be written in full form at the first appearance of the short form.

INRODUCTION:

#Author has used "." sign before the reference (e.g., ".(1-3)" in line 52). This must be revised. The full stop sign (.) must be used after the reference (i.e., "(1-3)."). This issue should be checked for the full manuscript.

METHODS:

#Line 84- "MAHMAZ situated the new...…." here, situated should be replaced by "established".

#Line 119- unnecessary gap should be removed.

#Line 222- "These costs are not are not included..." sentence should be corrected by removing "are not" for once.

RESULTS:

#The description and the value of the tables are not matching. The inconsistency was found in full result section. Either the tables or the descriptions have problems. This inconsistency must be met properly.

DISCUSSION:

# The whole discussion section should be checked for the standard of the English language that has been used in the manuscript.

7. PLOS authors have the option to publish the peer review history of their article (what does this mean?). If published, this will include your full peer review and any attached files.

**Do you want your identity to be public for this peer review?** For information about this choice, including consent withdrawal, please see our Privacy Policy.

Reviewer #2: No

Reviewer #3: No

---

## [Editor Report · Decision Letter 3]

3 Mar 2023

A cost description of the setup costs of community-owned maternity waiting homes in rural Zambia

PGPH-D-22-00186R3

Dear Ms Juntunen,

We are pleased to inform you that your manuscript 'A cost description of the setup costs of community-owned maternity waiting homes in rural Zambia' has been provisionally accepted for publication in PLOS Global Public Health.

Best regards,

Abdur Razzaque Sarker, PhD

Academic Editor